# Intercellular forces driving stratification in a two-layer corneal epithelium: Insight from a Voronoi cell-based simulation model

Neda Khodabakhsh Joniani[1]*, David Martinez-Martin[2,3], Peter S. Kim[1�], James Guy Lyons[4,5☯]

1 School of Mathematics and Statistics, University of Sydney, Sydney, New South Wales, Australia, 2 The University of Sydney Nano Institute (Sydney Nano), The University of Sydney, Sydney, New South Wales, Australia, 3 School of Biomedical Engineering, The University of Sydney, Sydney, New South Wales, Australia, 4 Centenary Institute, University of Sydney, Sydney, New South Wales, Australia, 5 Department of Dermatology, University of Sydney at Royal Prince Alfred Hospital, Sydney, New South Wales, Australia

☯ These authors contributed equally to this work.
* n.khodabakhsh@sydney.edu.au

## Abstract

The cornea is a self-renewing, multilayered tissue maintained with remarkable precision. Its outermost layer, the corneal epithelium, consists of five to seven stratified cell layers, sustained by two coordinated processes: the centripetal migration of transit amplifying cells (TACs) from peripheral limbal epithelial stem cells (LESCs), and delamination (vertical movement) of cells between layers. Despite this well-organized renewal, the mechanisms governing epithelial stratification remain largely unknown. In this study, we present a two-dimensional Voronoi cell-based model that captures the dynamics of epithelial stratification. Our model incorporates two distinct epithelial layers—the basal and the suprabasal layers—and accounts for key cellular processes. These processes are mediated by mechanical interactions such as cell-substrate adhesion, as well as horizontal and vertical intercellular forces.

Our simulations show that cell delamination, which drives stratification, is strongly linked to TAC proliferation. In contrast, LESC division remains largely unchanged, suggesting that TACs buffer LESC activity, consistent with the slow-cycling nature of stem cells. This reveals that processes weakening the cell-to-substrate interaction will enhance the turnover of epithelial cells without the need for external growth factor induction, which is a notable finding. Interestingly, while increased shedding promotes division and delamination, excessive shedding leads to mechanical compensation through cell stretching in the upper layers. This mechanical response provides a simple, plausible explanation for the presence of enlarged cells in the superficial epithelial layers, while not excluding the potential contributions of other mechanisms. Our model reveals a direct link between the shedding rate and the centripetal velocity of clonal growth, predicting that increased surface cell loss

**Data availability statement:** All relevant code used for running simulations, and plotting is available on a GitHub repository at https://github.com/Neda-Khodabakhsh/corneal_epithelium_code_data.

**Funding:** PSK (Peter S. Kim) gratefully acknowledges support from the Australian Research Council Discovery Project DP230100485. https://www.arc.gov.au/funding-research/funding-schemes/discovery-program. N.Kh.J (Neda Khodabakhsh Joniani) gratefully acknowledges support from the Postgraduate Research Scholarship in Mathematics and Statistics. No. SC4238 The funders had no role in study design, data collection and analysis, decision to publish, or preparation of the manuscript.

**Competing interests:** The authors have declared that no competing interests exist.

accelerates cell movement-a response similar to wound healing, where cells rapidly migrate to restore the damaged area.

These results highlight how cell size, migration, and turnover are tightly coupled, and offer deeper insights into how physical forces work together to maintain and rapidly restore epithelial integrity. Although the real cornea contains five to seven layers, this two-layer framework focuses on the key mechanical principles of stratification and can be viewed as a foundational step toward more comprehensive multilayer modelling.

## Author summary

The cornea is the eye's clear outer layer, helping to protect the eye and focus light for sharp vision. Its outermost surface is made up of several layers of cells that are constantly renewed to keep the tissue healthy. This renewal happens through two main processes: new cells are produced by stem cells on the edge of the cornea and move inward, while older cells gradually move upward through the layers and eventually shed from the surface. However, the exact way these movements and layers are controlled is still not fully understood. In this study, we developed a computer model to explore how these cells grow, move, and replace one another over time. The model includes the physical forces, such as how strongly cells stick to each other and to the surface beneath them. Our results show that the upward movement of cells plays an important role in how often cells divide. We also found that when surface cells are lost more quickly such as in injury the cells below respond by dividing and moving faster to repair the surface. This helps us better understand how the cornea repairs itself and maintains its protective surface.

## 1 Introduction

The cornea, due to its anatomical simplicity, optical clarity and accessibility for imaging, has been of great interest to many researchers from biological and theoretical views. Loss of normal transparency of the cornea or its geometrical structure causes a group of corneal disorders, such as corneal opacity and keratoconus [1,2].

The cornea consists of several layers, each contributing to its overall function and health [3]. Among these layers, the corneal epithelium, the top layer, is essential for maintaining the cornea's integrity and optimal visual performance [4].

Epithelial tissues constitute one of the most significant types of body tissue, serving to line both the inner and the outer surfaces of organs. This tissue is composed of interconnected cells exhibiting diverse morphologies and functions. Furthermore, epithelial cells can be organized in a single layer or in multiple layers, depending on their specific functions [5].

The corneal epithelium, which contributes to corneal transparency and protection, is a multilayer, or "stratified" epithelium, comprising the innermost basal layer, followed by the suprabasal (wing) and superficial cell layers [4]. In contrast to simple epithelia, which facilitate the rapid transport of molecules across them,

stratified epithelia serve primarily as barriers to most molecules and to protect underlying tissue from microorganisms and ultraviolet radiation [6].

For epithelial stratification to occur, cells move upward from the basal layer to the layer above and up subsequent layers through a process known as delamination [7]. In addition, corneal epithelial cells can migrate within a layer horizontally. As such, vertical and horizontal movements are the two main orientations for epithelial cell movement [8]. In the corneal epithelium, cell proliferation normally only occurs in the basal layer [9].

An explanation for how the corneal epithelium is repopulated during homeostasis is offered by the XYZ hypothesis. In this theory, X represents the proliferation and anterior migration of basal epithelial cells, Y denotes the centripetal movement of epithelial cells towards the center of the cornea, and Z refers to the loss of cells from the corneal surface through shedding [10].

A number of studies have focused on corneal transparency and shape problems using differential equations and numerical methods, by providing mathematical analysis to understand the physical characteristics of the cornea [11–16]. Moreover, some models use Zernike polynomial approximations to investigate corneal or lens aberrations [17]. Zernike polynomials are orthogonal mathematical functions used to describe optical aberrations on circular surfaces like the cornea or lenses. Defined over a unit disk, they combine radial and angular components, making them ideal for modelling and quantifying distortions in curved optical systems with high precision.

Stochastic and reaction-diffusion models proposed for exploring the process of corneal epithelium homeostasis and wound healing, providing insights into the dynamic interactions between cells and the wound healing speed [18–21].

In 2016, Lobo et al. [22] used a Voronoi cell-based model to show that corneal epithelial cells in the basal layer can self-organize into cohesive, centripetal growth patterns through local cell-to-cell interactions without the need for chemotaxis or other nonlocal signalling.

However, as previously noted, the corneal epithelium is stratified, and questions remain as to how the upper epithelial layers might regulate the organization and turnover of the corneal epithelium.

Understanding the mechanical structure and dynamic behaviour of epithelial cellular networks poses significant mathematical challenges [23]. These arise primarily from the complex interplay of internal and external forces that act simultaneously within the tissue. Moreover, biological processes such as cell growth, division, and delamination occur concurrently, further increasing the complexity of the system. Together, these mechanical interactions and cellular events create a highly dynamic and nonlinear environment that is difficult to capture mathematically and computationally. To address these challenges and develop a realistic model of corneal epithelial cell behavior, it is essential to integrate the mechanical interactions and biological mechanisms across the different epithelial layers by considering forces that arise from cell population density, extracellular matrix stiffness, and tissue architecture.

In this study, we developed a Voronoi cell-based model (VCBM) accounting for the basal and the second (suprabasal) layers of the corneal epithelium. The focus is on understanding how the dynamics of the XYZ hypothesis works and providing insight into the mechanisms that regulate epithelial stratification during homeostasis.

The proposed model incorporates biophysical forces acting within and between cells and layers, thereby providing a more comprehensive representation of epithelial behavior. In contrast to the previous model [22], which was limited to simulating cell growth and division within a single layer, the current framework features multiple two-dimensional layers and includes cell delamination component and forces between the layers. This simplified two-layer model captures the fundamental mechanics of stratification and serves as a stepping stone toward multilayer representations of the cornea.

## 2 Corneal structure

The cornea is the outermost transparent layer of the eye, serving as a window that facilitates the focusing of light and provides protection to the internal structures of the eye. The human cornea is arranged in five basic

layers – epithelium, Bowman's layer, stroma, Descemet's membrane, and endothelium, each with unique structural and functional properties [24].

**Corneal epithelium.** The exterior surface of the cornea is the epithelium. The corneal epithelium is stratified and can have up to seven layers of cells [3]. From the basal layer, epithelial cells develop into one layer of suprabasal (wing) cells and followed by 3 to 5 flat post-mitotic superficial cell layers (Fig 1) [25]. The cells become wider in the plane of the cells and thinner in height as they move vertically through the layers.

**Limbal epithelial stem cell (LESC) and transit amplifying cell (TAC).** An important component is the limbus, a ring forming a border between the cornea and the sclera (the white outer region of the eyeball), which is the residence of limbal epithelial stem cells (LESCs) (Fig 1) [26]. LESCs, like other stem cells, are unspecialized cells with high proliferative capacity, slow cycling and long lifespans. They are regarded as the source of replenishment of corneal epithelial cells [25]. LESCs are believed to divide asymmetrically every 4-8 days on average [27] to produce an LESC and a TAC in the basal layer [28]. The resulting LESC stays in the limbus and the resulting TAC migrates centripetally to form part of the basal corneal epithelium [29]. TACs divide symmetrically to produce two TACs in the basal layer approximately every 3.5 days [27]. Unlike LESCs, TACs have a finite proliferative potential. It is worth mentioning that there is also evidence that LESCs can divide symmetrically to produce two LESCs to replenish the limbus or two TACs [30,31].

It should be noted that the death rate of TACs in the basal layer is normally almost zero and TACs primarily turnover by moving upward through the delamination process and then are shed from the surface of the cornea [32]. Hence, for simplicity, in our model we assume that the death rates for TACs and LESCs are both zero.

**XYZ hypothesis.** Corneal epithelial cells are constantly moving. The primary directions for epithelial cells are centripetal migration [8] within a layer and vertical movement (delamination) to an upper layer. Cells are continuously lost from the superficial layer through the shear forces of eyelid blinking and eye rubbing. The structure of the corneal epithelium is preserved, not only by basal epithelial cell division, but also by migration of new basal cells into the cornea from the limbus (Fig 1). Therefore, cell renewal must balance cell loss to maintain the corneal epithelium, which is represented as the X, Y, Z hypothesis, proposed by Thoft and Friend in 1983 [10]. In this theory, X represents proliferation, Y is the migration of new epithelial cells into the base of the cornea from the limbus, and Z is epithelial cell loss (Fig 1). They proposed that $X + Y = Z$, suggesting that disruptions in any of the components X, Y, or Z may contribute to the development of corneal epithelial disorders.

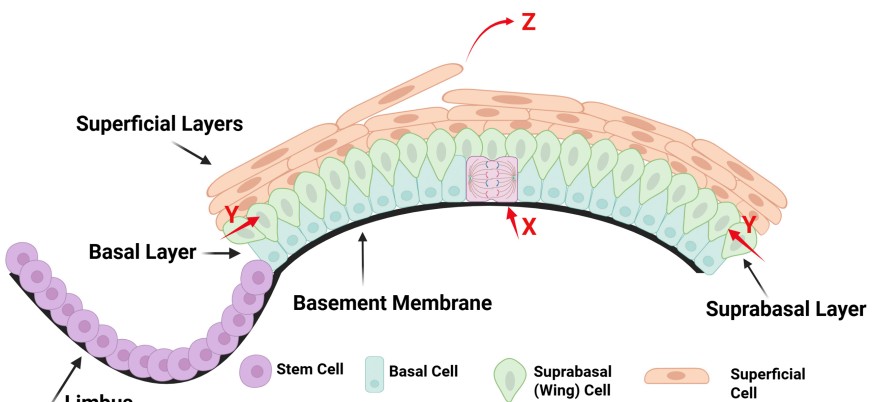

**Fig 1**. **The corneal epithelium is multilayered and comprises basal, suprabasal, and superficial cells, typically organized into 5 to 7 layers.** X, Y and Z describe the population balance of corneal epithelial cells. Limbus circumscribes the cornea and is the residence of LESCs. This figure is created in Biorender (https://BioRender.com/0mvrjkg).

## 3 Model description

Our model is based on a VCBM, which is a type of agent-based model based on points called Voronoi centres. Cell regions are determined by the set of points closest to each Voronoi centre, resulting in a tiling of the plane called a Voronoi tessellation [33]. One of the key features of a VCBM is that cells are not limited to a lattice and are free to deform and move throughout the space [34–36]. At each time step $\Delta t$, our model is updated according to three processes: cell movement, cell division and cell delamination.

### 3.1 Cell movement

Following the formulation of Jenner et al. [37,38] for cell movement, we imagine that a cell has connections with its neighbours via springs and its movement depends on the spring forces obtained from Hooke's Law (Fig 2). The displacement vector of cell $i$ is derived from Newton's second law of motion

$$m_i \frac{d^2 \vec{r}_i}{dt^2} = \sum_{j \neq i} \vec{F}^I_{i,j} + \vec{F}^V_i, \tag{1}$$

where $m_i$ is the mass of cell $i$, $\vec{r}_i$ is its position, $\vec{F}^I_{i,j}$ is the spring force on cell $i$ from cell $j$ with the spring constant $\mu$, and $\vec{F}^V_i$ is the viscous drag force acting on cell $i$. $\vec{F}^V_i$ is generated by moving through a viscous fluid and is proportional to the velocity of the cell with the constant $\eta$. Due to the strong friction in the environment, we assume cell movements are overdamped and so

$$m_i \frac{d^2 \vec{r}_i}{dt^2} \sim 0.$$

Hence, Eq 1 can be simplified as

$$\sum_{j \neq i} \vec{F}^I_{i,j} = -\vec{F}^V_i = \eta \vec{v}_i,$$

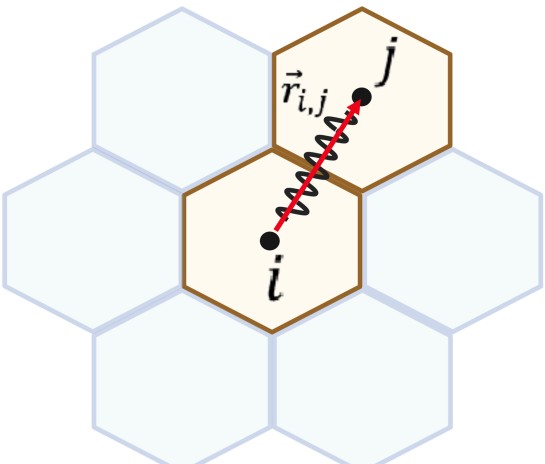

**Fig 2**. **Spring force acting on cell *i* from its neighbour *j*.** The force is calculated from Hooke's law, where $\vec{r}_{i,j}$ is the vector from cell *i* to cell *j*. This figure is created in Biorender (https://BioRender.com/v227r7h).

where $\vec{v}_i$ is the cell's velocity. Approximating velocity over small $\Delta t$, we obtain

$$\sum_{j \neq i} \vec{F}^I_{i,j} \sim \eta \frac{\vec{r}(t + \Delta t) - \vec{r}(t)}{\Delta t}.$$

Thus based on Hooke's Law, the displacement $\vec{r}_i(t + \Delta t)$ is

$$\vec{r}_i(t + \Delta t) = \vec{r}_i(t) + \frac{\Delta t}{\eta} \sum_{j \neq i} \vec{F}^I_{i,j} = \vec{r}_i(t) + \lambda \Delta t \sum_{j \neq i} \frac{\vec{r}_{i,j}(t)}{\|\vec{r}_{i,j(t)}\|} \left( s - \|\vec{r}_{i,j}(t)\| \right), \tag{2}$$

where $\vec{r}_{i,j}$ is the vector from cell $i$ to cell $j$, $\|\vec{r}_{i,j}(t)\|$ is the length of the vector $\vec{r}_{i,j}$, $s$ is the spring rest length, and $\lambda = \mu/\eta$ is the ratio between the spring constant $\mu$ and the damping constant $\eta$. $\lambda$ influences the velocity of the relaxation process [33].

## 3.2 Cell division

As we explained in Sect 2, in our model, LESCs divide asymmetrically to produce one LESC and one TAC in the basal layer. The resulting LESC stays in the limbus and the resulting TAC is placed a slight distance of 0.05 $\mu$m from the parent LESC in a random direction towards the interior of the cornea. We assume that TACs divide symmetrically to a maximum number of divisions $n_{max}$. When a TAC's number of divisions reaches $n_{max}$, it becomes a terminally differentiated cell (TDC) and no longer divides. We also assume that newly divided LESCs and TACs undergo cell cycles of duration $T_{LESC}$ and $T_{TAC}$, respectively, and cannot divide again until this time has elapsed.

Cell divisions are also governed by local population densities, measured by what we call a neighbour force. Experiments of Cattin et al. [39] showed that cell division slows and stops as more external force is applied. To have a mathematical condition for this, we introduce a neighbour force $F^{neigh}_{i,j}$ as the spring force cell $i$ receives from cell $j$:

$$F^{neigh}_{i,j} = c_{neigh}(s - \|\vec{r}_{i,j}(t)\|), \tag{3}$$

where $c_{neigh}$ is the neighbour force constant.

At each time step, an LESC or a TAC will divide if both of the following conditions hold:

- The minimum cell cycle time for LESCs and TACs ($T_{LESC}$ and $T_{TAC}$, respectively) has elapsed since the last division.
- The maximum neighbour force on cell $i$ is below a threshold, i.e., $\max_{j \neq i} F^{neigh}_{i,j} < F^{th}$, where $F^{th}$ is some threshold.

## 3.3 Cell delamination

By adding a layer above the basal layer, we need to decide how forces will be transmitted between layers, which will determine whether a cell delaminate or not. To maintain a cell population balance, when a cell in the basal layer is in an area of high cell density and the area in the layer above is a low cell density area, then the cell should have a greater probability of delaminating. This is analogous to the population pressure regulation of horizontal movement.

In this scenario, we model that cell delamination is regulated by three forces: the cell-substrate adhesion force, the horizontal force, and the vertical force. These forces interact with the varying densities of surrounding cells, influencing the likelihood of delamination based on the environment.

**3.3.1 Cell-substrate adhesion force.** The cell-substrate adhesion force is a traction force that cells in the basal layer of the epithelium experience as they adhere to the underlying basement membrane. This attraction attaches the basal layer to the underlying structure (Fig 3) and is tightly related to many cellular mechanisms and functions, including cell motility and proliferation [40].

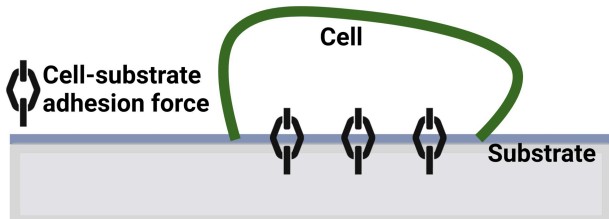

**Fig 3**. **Schematic for the cell-substrate adhesion force between the cell in the basal layer and the substrate.** This figure is created in Biorender (https://BioRender.com/al1x5yk).

In our model, this force is proportional to the cell surface area,

$$F_i^{\text{cell-substrate}} = c_{\text{cell-substrate}}\, A_i,$$

where $c_{\text{cell-substrate}}$ is a constant and $A_i$ is the surface area of cell $i$.

**3.3.2 Horizontal force.** The horizontal force is a subset of spring forces imparted on a cell by its neighbors within the same layer of an stratified epithelium. Put differently, a cell feels tension or pressure from its neighboring cells and responds by moving appropriately (Fig 4A). This force has a crucial role in cell migration, delamination and proliferation within an epithelial tissue [41,42].

In our model, this force is defined as the sum of all spring forces directed towards a cell leading to the stretching or compression of the cell promoting its retention within its layer or its delamination, respectively (Fig 4A).

$$F_i^{\text{horizontal}} = \sum_{i \neq j} c_{\text{horizontal}}(s - \|\vec{r}_{i,j}(t)\|),$$

where $c_{\text{horizontal}}$ is the horizontal force constant.

**3.3.3 Vertical force.** The vertical force arises between the epithelial layers. In our model, we consider two layers, and the vertical force results from cells in the second layer acting on the basal cells below (Fig 4B). To calculate the vertical

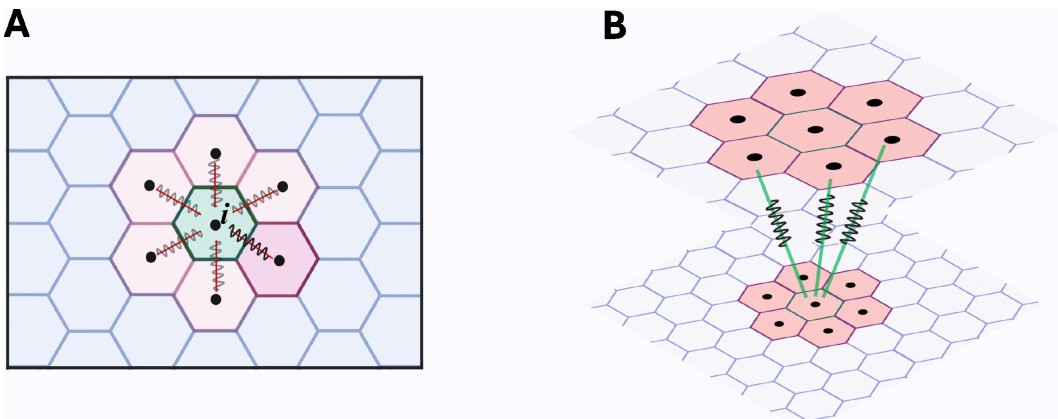

**Fig 4**. **Schematic representation of different types of spring forces in our model.** (A) is the horizontal force on a cell from its neighbors. (B) is the vertical force from cells in the second layer that overly the basal cell. This figure is created in Biorender (https://BioRender.com/erb0a3d).

force for a specific basal cell $i$, we consider the cells in the second layer that are above cell $i$. Firstly, we find cells in the second layer whose Voronoi centres are within one average cell diameter $d$ of cell $i$. Hence, within these cells, if $(x_i, y_i)$ are the coordinates of the centre of basal cell $i$, then all of the cells $j$ in the second layer whose $(x,y)$-coordinates are such that

$$(x_j - x_i)^2 + (y_j - y_i)^2 < d^2 \quad (d = \text{average cell diameter}),$$

are our desired cells (Fig 5A).

As such, the distance between cell $i$ and these cells are

$$d_{i,j} = \sqrt{(x_j - x_i)^2 + (y_j - y_i)^2 + h^2},$$

where $h$ is the distance between two layers. Therefore, the vertical force is computed as the summation of the vertical components of all the spring forces acting on the basal cell $i$.

Therefore, our vertical force is computed as the summation of all spring forces acting on the basal cell.

$$F_i^{\text{vertical}} = \sum_{j \neq i} \frac{h}{d_{i,j}} \, c_{\text{vertical}}(d_{i,j} - s),$$

where $c_{\text{vertical}}$ is the vertical force constant. The factor $h/d_{i,j}$ multiplied by the spring force is the $\sin(\theta_j)$ term in the vertical component, where $\theta_j$ is the angle between cell $i$ in the basal layer and cell $j$ in the second layer (Fig 5B).

Our condition for cell delamination is

$$F_i^{\text{horizontal}} + F_i^{\text{vertical}} > F_i^{\text{cell-substrate}}. \tag{4}$$

### 3.4 Parameter estimation

All cell parameters and their values are collected in Table 1.

TACs have a limited number of symmetrical cell divisions ($n_{\text{max}}$) to produce two TACs, which we estimate to range from 2 to 4 [9]. Di Girolamo et al. [45] observed that the radius of a flat-mounted mouse cornea is around 100 cells and

**A**

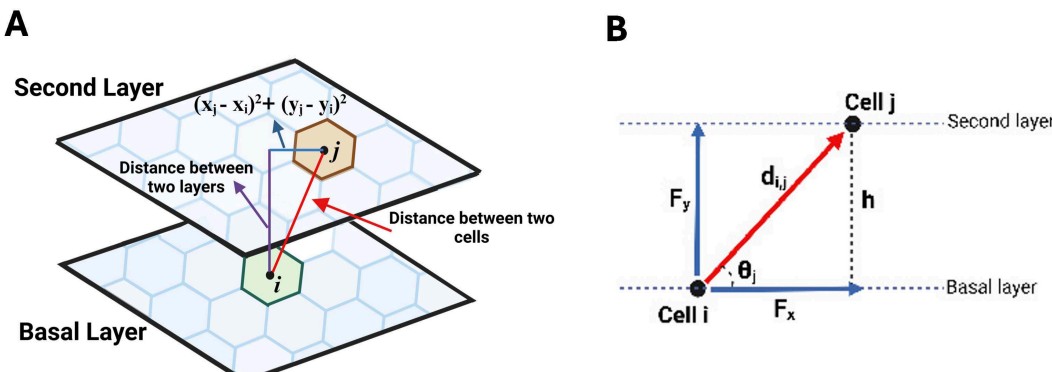

**B**

**Fig 5**. **Schematic representation of vertical force on the basal cell.** (A) The vertical force on the basal cell is considered as the summation of all the spring forces from the cells in the second layer that are above the basal cell. (B) is the vertical component of the force from cells in the second layer on basal cells. This figure is created in Biorender (https://BioRender.com/33sg6if).

**Table 1**. Model parameters and assumed values.

| Parameters | Value | Description | Reference |
|---|---|---|---|
| $n_{max}$ | 2-4 | Maximum number of TAC divisions | [9] |
| $s$ ($\mu$m) | 15 | Spring rest length | Assumed |
| $h$ ($\mu$m) | 15 | Distance between two layers | [22] |
| $\lambda$ (hour$^{-1}$) | 0.055 | Movement parameter for the Voronoi model | Assumed |
| $T_{LESC}$ (hour) | 48 | Minimum time between cell divisions for LESCs | [27] |
| $T_{TAC}$ (hour) | 24 | Minimum time between cell divisions for TACs | [9] |
| $F_{TAC}^{th}$ (pN) | 3 | Force threshold for TAC divisions | [41] |
| $F_{LESC}^{th}$ (pN) | 5 | Force threshold for LESC divisions | [41] |
| $c_{cell\text{-}substrate}$ (pN/$\mu$m$^2$) | 0.005 | Cell-substrate adhesion force constant | Assumed |
| $c_{horizontal}$ (pN/$\mu$m) | 0.1 | Horizontal force constant | Assumed |
| $c_{vertical}$ (pN/$\mu$m) | 0.7 | Vertical force constant | Assumed |
| $c_{neigh}$ (pN/$\mu$m) | 1 | Neighbour force constant | Assumed |
| $r_{shed}$(/hour) | 0.0005 | Shedding rate from the second layer | [43] |
| Velocity ($\mu$m/hour) | 0.45 | Centripetal velocity of clonal growth | [44] |
| $R$ ($\mu$m) | 500 | Radius of the disc | Assumed |
| $N_{LESC}$ | 50 | Number of LESCs in the limbus | Assumed |
| $\Delta t$ (hour) | 5 | Time step | Assumed |

1500 $\mu$m, so the average diameter of the TACs is 15 $\mu$m [22]. As we will see in the sensitivity analysis (Fig 15E, 15F), number of cells and subsequently cell diameter in the basal and second layers are sensitive to the rest length $s$, so to get a cell dimeter around 15 $\mu$m in the basal layer, we choose $s = 15$ $\mu$m. In Sect 4.3, we demonstrate that as we progress to the upper layers, the cell size increases as the cells flatten out. Consequently, we assume the rest length in the second layer to be approximately 20$\mu$m. For convenience, we estimate the distance between layers ($h$) to be the same as the average diameter of a TAC in the basal layer, i.e., $h = 15$ $\mu$m.

The movement parameter for the Voronoi model is described by $\lambda$ in Eq 2, which is set to $\lambda = 0.055$ hour$^{-1}$. A small value of $\lambda$ compresses cells on the rim, whereas larger values result in unrealistically rapid cell movement, thereby disrupting centripetal movement. Simulation results indicate that minor deviations from this value do not produce any significant effects on the overall simulation outcomes.

The parameters $T_{LESC}$ and $T_{TAC}$ represent the minimum times between cell divisions for LESCs and TACs, respectively. Based on experimental evidence, these minimum division times are around 48 hours for LESCs and 24 hours for TACS [9,46].

Force-related parameters $c_{cell\text{-}substrate} = 0.005$ pN/$\mu$m$^2$, $c_{horizontal} = 0.1$ pN/$\mu$m, $c_{vertical} = 0.7$ pN/$\mu$m were chosen to obtain the desired dynamics; however, in the sensitivity analysis in Sect 5 we vary them within a range of $\pm 50\%$ of the base values to see their effect on the output of the model. Gudipaty et al. [41] reported that epithelial cells increase their division rate when cell density decreases by approximately 1.4-2.5-fold. In our model, considering a corneal diameter of 1000 $\mu$m and an average basal cell diameter of approximately 15 $\mu$m, the basal layer contains roughly 4400 cells. A 1.4-fold reduction in cell density corresponds to an increase in cell diameter to about 17.8 $\mu$m, implying an extension of approximately 2.8 $\mu$m from the rest length in the neighbor force (Eq 3) for most cells. With $c_{neigh} = 1$ pN/$\mu$m, this results in a neighbor force of $F^{neigh} = 2.8$ pN. Accordingly, we set the threshold force for TAC division to $F_{TAC}^{th} = 3$ pN. We assigned a slightly higher threshold $F_{LESC}^{th}$ to promote sufficient TAC production within a shorter time period. Interestingly, despite the higher $F_{LESC}^{th}$ compared to that of TACs, the model exhibits an emergent behaviour in which the division rate of LESCs remains lower than that of TACs.

As mentioned before, Di Girolamo et al. [45] observed that the radius of a mouse cornea is 1500 $\mu$m and Dora et al. [47] estimated that a mouse limbus has around 1000 LESCs, but for computational efficiency, we usually simulate the

system for radius $R = 500$ $\mu$m and $N_{\text{LESC}} = 50$ LESCs. Nontheless, we conduct one simulation for a realistically sized mouse cornea with $R = 1500$ $\mu$m and $N_{\text{LESC}} = 1000$ (S1 Fig).

Ren et al. [43] reported that the cell shedding rate of the in vivo rabbit corneal epithelium ranges from 5 to 15 cells per minute per cornea. Given that the diameter of the rabbit cornea is approximately 13 mm and the diameter of our simulated cornea is 1 mm, the size of our simulated cornea is roughly 169 times smaller than that of the rabbit's cornea. Consequently, the shedding rate for our cornea is calculated as follows:

$$5 \text{ cells/min} \times (1/169) = 0.0296 \text{ cells/min},$$

which is equivalent to 1.7760 cells per hour. For a diameter of 1000 $\mu$m and a second layer cell diameter around 17-18 $\mu$m [3], we estimate that there are approximately 3500 cells in the second layer. Therefore, the shedding rate from the second layer ($r_{\text{shed}}$) is approximately

$$\frac{0.0296 \text{ cells/min}}{3500 \text{ cells}} = 0.0005/\text{hour}.$$

We update the system with a time step $\Delta t = 5$ hours. The centripetal pattern was preserved for $\Delta t \leq 5$ but deteriorated for larger $\Delta t$. This limitation arises from the discrete-time nature of the agent-based update rules: larger time steps reduce accuracy and violate the assumptions underlying event probabilities and motion updates.

Hence, we find that this time step is small enough to capture the relevant dynamics without being too small as to make the simulations computationally too demanding.

## 3.5 Sensitivity analysis

To enhance our comprehension of the impact of parameter variations on centripetal growth, division rates of LESCs and TACs, cell sizes in each layer, and delamination rate, we conducted a global sensitivity analysis. Utilizing Latin Hypercube Sampling (LHS) [48], we generated 300 samples from the parameter space within a range of $\pm 50\%$ of the base values presented in Table 1. Subsequently, we simulated our model and computed the Spearman rank correlation coefficient (SRCC) for each parameter to assess its influence on centripetal movement, cell division rates, cell diameter and delamination rate.

## 3.6 Initial conditions

For simplicity, our model depicts the cornea as a flat circle instead of a spherical cap. As such, the ratio of the TACs to LESCs is slightly lower than the real geometry of the cornea. In the basal layer, we distribute the centres of our LESCs evenly around the rim. In other words, if we have $N_{\text{LESC}}$ LESCs around a disc of radius $R$ centred at the origin of the $xy$-plane, the Voronoi centres for the LESCs will be located at $(R \cos(2\pi i/N), R \sin(2\pi i/N))$ for $i = 0, \dots, N_{\text{LESC}} - 1$. Unlike the basal layer, in the second layer we do not have LESCs. Instead, we place Voronoi centres at the same locations as the LESCs in the basal layer as fixed, inert "phantom" cells at the boundary to prevent other cells from moving beyond the disc.

We simulated the system starting from empty basal and second layers for up to 5,000 hours ($\approx 208$ days), until it reached a stable pattern as shown in Fig 6. Then, we used these stable states as initial conditions for further simulations. In our simulations, LESCs are displayed as the coloured points on the disc's edge in the basal layer and cell colours inside the disc correspond to different LESC lineages. Our colour coding enables us to observe the centripetal growth pattern of LESC lineages.

PLOS Computational Biology

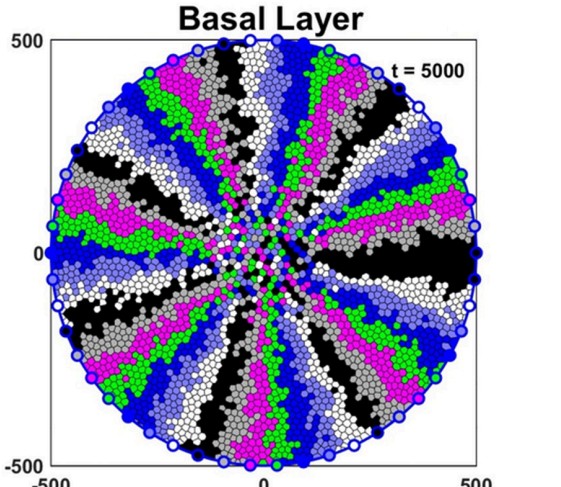 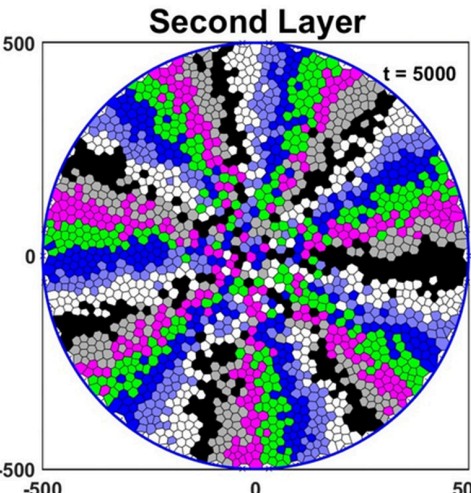

**Fig 6**. Simulations commence with the basal and second layers initially empty, progressing through 5,000 hours until a stable pattern is reached. LESCs are displayed as the coloured points on the disc's edge in the basal layer and cell colours inside the disc correspond to different LESC lineages. The radius of the cornea in our simulations is 500 $\mu$m.

## 4 Results

### 4.1 Centripetal growth in the basal and the second layers

Fig 7A, 7B and 7C are the simulation of the basal and the second layers for 15000 hours and $n_{max}$ ranging from 2 to 4, as reported in [9]. We consider $n_{max} = 1$, $n_{max} = 5$ in (S2 Fig) to see how the centripetal pattern changes for higher and lower values. Fig 7D is a fluorescence micrograph of a cornea from a K14CreERT2-Confetti mouse adopted from [22].

It can be seen that the spoke-like pattern in the basal and the second layers is preserved, which is consistent with fluorescence micrograph of mouse corneas (Fig 7D).

To compare growth in the basal and the second layers for different $n_{max}$, we considered the linear displacement (LD) [22], which is a measure of the degree to which migration is centripetal. To determine the LD of the clonal cell lineages in the basal layer, we calculated the perpendicular distance between the specific cell and the line connecting its originating LESC to the centre. We have a perfect centripetal movement when the LD is zero, and we get less cohesive centripetal movement as the LD increases. For a particular LESC and its lineage, we calculated the mean LD for cells for 20 simulations, and to compare the differences in clonal groups of cells with different $n_{max}$, we performed a Kruskal-Wallis test with Dunn's multiple comparisons test (Fig 8). As $n_{max}$ increases, the cell clonal cohesion in the basal layer decreases. This can also be seen qualitatively in Fig 7. As such, in the basal layer $n_{max} = 2$ and $n_{max} = 4$ has the highest and the lowest spoke-like patterns, respectively. For the second layer, we used the phantom cells situated at the periphery of the cornea, directly above the corresponding LESCs, as reference points to compute LDs for cell lineages. As illustrated in Fig 8, the second layer maintains its centripetal pattern as $n_{max}$ increases to 4. Moreover, we can see that the centripetal pattern in the second layer is almost independent of $n_{max}$. However, a slight increasing trend remains evident for $n_{max} = 3$ and 4. Given that cell division occurs exclusively in the basal layer, it appears that the pattern in the second layer remains unaffected by variations in $n_{max}$.

### 4.2 Relationship of cell division and delamination rates to $n_{max}$

Fig 9 shows cell division and delamination rates relative to $n_{max}$. Fig 9A illustrates that LESCs divide slower on average than TACs and that the relative difference increases as $n_{max}$ increases, which confirms the distinct slow-cycling

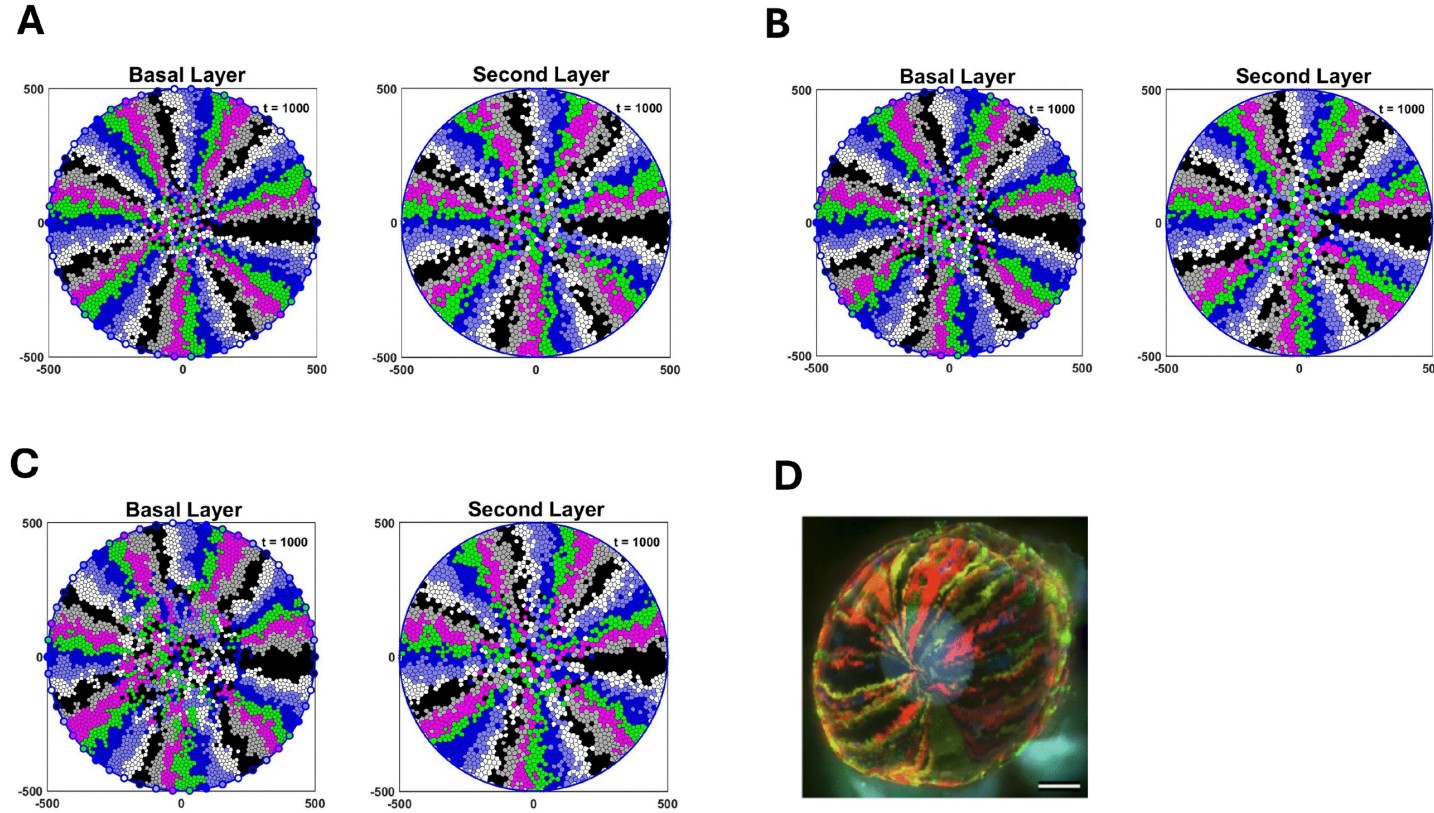

**Fig 7**. **Representative model simulations for the basal and the second layers over 15000 hours for (A)** $n_{max} = 2$, **(B)** $n_{max} = 3$ **and (C)** $n_{max} = 4$. The radius of the cornea in our model is assumed to be 500 $\mu$m. (D) A fluorescence micrograph of a cornea from a K14CreERT2-Confetti mouse. Scale bar: 500 $\mu$m. This figure is adapted with permission from [22].

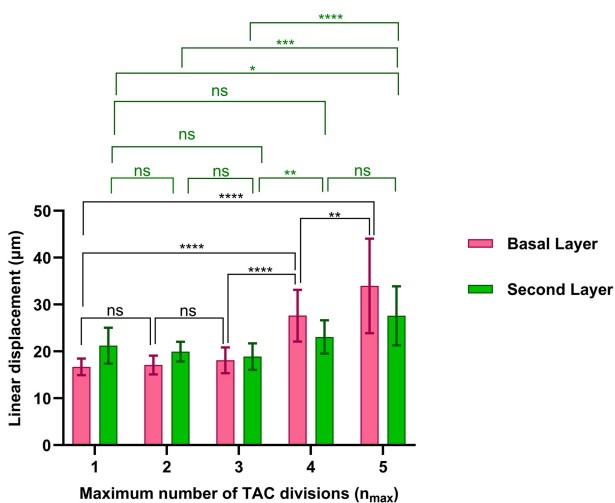

**Fig 8**. **As $n_{max}$ increases, the cell clonal cohesion in the basal layer decreases, whereas the centripetal pattern in the second layer is almost independent of $n_{max}$.** For each $n_{max}$, we started the simulation with the initial states shown in Fig 6 and continued for an additional 5000 hours. The LDs are calculated for the north-most LESC ($\theta = \pi/2$). The mean $\pm$ SD of the LDs are shown for 20 simulations for each $n_{max}$. Statistical differences were determined using Kruskal-Wallis test followed by Dunn's multiple comparisons test ("ns" means "not-significant", $^*p<0.05$, $^{**}p<0.01$, $^{***}p<0.001$, $^{****}p<0.0001$).

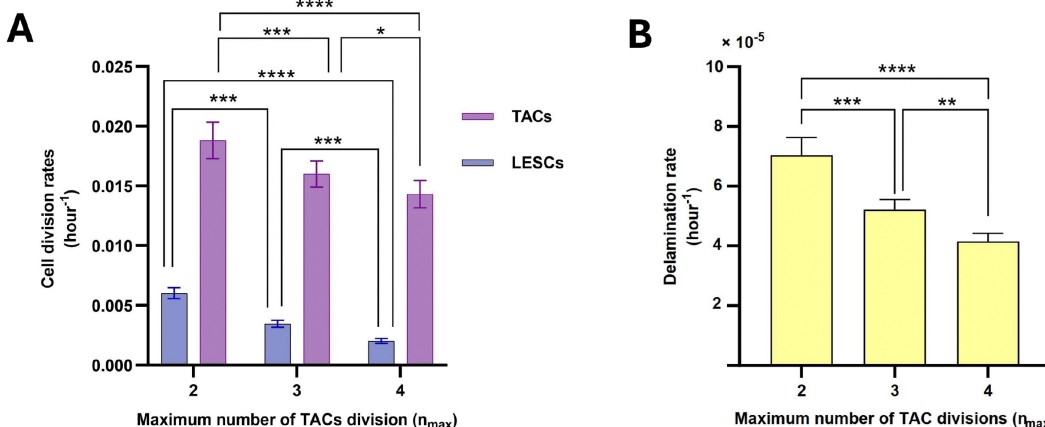

**Fig 9**. **Relationship of maximum number of TAC divisions, $n_{max}$, to cell division rates (A) and delamination rate (B).** As $n_{max}$ increases, cell division and delamination rates decrease. The mean $\pm$ SD of cell division and delamination rates are shown for 20 simulations for each $n_{max}$. Statistical differences were determined using Kruskal-Wallis test followed by Dunn's multiple comparisons test ($^{*}p<0.05$, $^{**}p<0.01$, $^{***}p<0.001$, $^{****}p<0.0001$).

characteristic of LESCs compared to TACs. As $n_{max}$ increases, LESCs exhibit a reduced division rate, since TACs undergo a greater number of divisions, diminishing the necessity for LESCs to divide to maintain a stable equilibrium of cell numbers.

Fig 9B shows that the delamination rate decreases as $n_{max}$ increases, which is consistent with the reduction in the division rate observed in Fig 9A. When $n_{max} = 2$, the division rate is at its highest; therefore, to maintain the population balance, a greater number of cells undergo delamination. Consequently, a decreasing relationship between $n_{max}$ and delamination rate is expected.

### 4.3 Relationship of delamination and shedding rates to LESC and TAC division rates

**Cell division and delamination rates.** To examine the effect of increasing delamination rate from the basal layer to the second layer on division rates, we imposed a gradual increase in delamination by reducing the cell-substrate adhesion force constant, $c_{cell\text{-}substrate}$. This could occur, for example, as a result of cell signaling or mutatgenesis [6].

As the delamination rate increases from $5.081 \times 10^{-5}$/hour to $6.460 \times 10^{-5}$/hour, TAC division rate exhibits a proportionate increase (Fig 10A). This is a particularly important observation, demonstrating that reduced cell-substrate adhesion promotes epithelial cell turnover independently of growth factor signaling. In addition, Fig 10B illustrates that an increase in the delamination rate results in a greater number of cells in the second layer, leading to a significant reduction in the mean cell diameter in the second layer. It should be noted that the loss of cells in the basal layer results in only a slight increase in the mean cell diameter of the basal layer, as cell division can compensate for the loss of basal cells.

For delamination rates of $6.263 \times 10^{-5}$/hour and beyond, only marginal changes in the TAC division rate were observed (Fig 10A). Moreover, within this range of delamination rates, Fig 10B reveals the cell size in the second layer decreases slightly, with no significant change in the basal cell size. Fig 10A illustrates that the LESC division rate exhibits only minimal variation as the delamination rate increases.

**Relationship of cell division rates to shedding rate**. In our model, as the shedding rate from the second layer increases, the number of cells in that layer decreases, resulting in a reduction of the push-down force from $F^{vertical}$. This results in an increased vertical force $F^{vertical}$ in the delamination condition in Eq 4 across the epithelium. This leads to a faster delamination rate, which in turn creates a lower cell density and more frequent high tension spring forces that induce basal cells to undergo division. As a result, an increase in the shedding rate induces a rise in the delamination rate

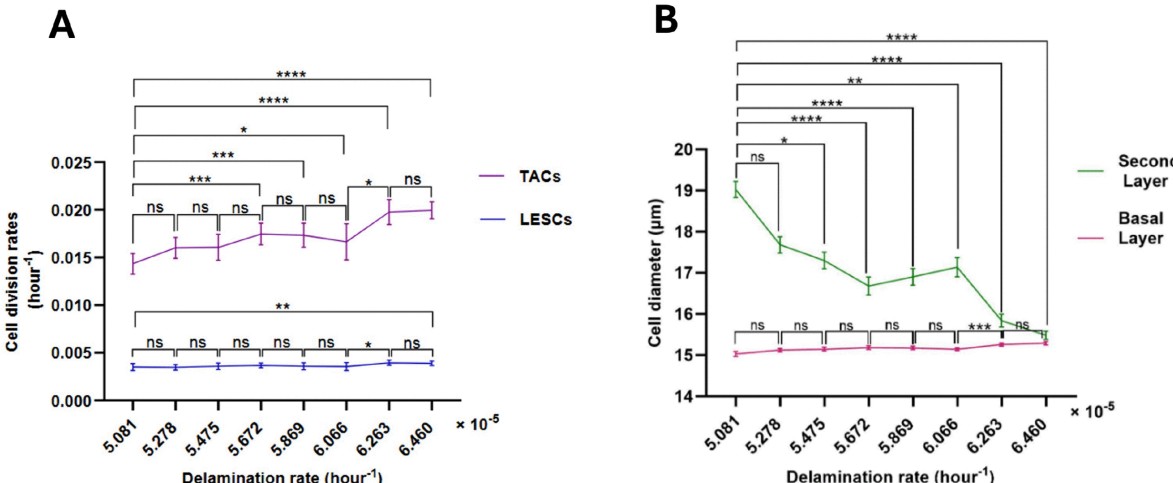

**Fig 10**. Relationship between the basal cell delamination rate and (A) cell division rates, and (B) cell size in the basal and the second layer. The mean ± SD of cell division rates and cell diameter are shown for 20 simulations for each delamination rate. Statistical differences were determined using Kruskal-Wallis test followed by Dunn's multiple comparisons test ("ns" means "not-significant", $^*p < 0.05$, $^{**}p < 0.01$, $^{***}p < 0.001$, $^{****}p < 0.0001$).

and division rates of LESCs and TACs (see Fig 11A, 11B). Moreover, as illustrated in Fig 11C, an increase in the shedding rate leads to a simultaneous enlargement of cells in both the basal and the second layers.

Notably, Fig 11B shows that, for $r_{shed} = 0.0016$/hour and higher values, delamination rate of increase slows down. This might arise because, for approximately $r_{shed} = 0.0016$/hour and beyond, as the basal cell area expands (see Fig 11C), the cell-substrate adhesion force, $F^{cell\text{-}substrate}$, exceeds the sum of the horizontal and vertical forces, $F^{horizontal} + F^{vertical}$ in Eq 4. Consequently, fewer cells meet the delamination condition, resulting in minimal changes to the rates of delamination and cell division. In this scenario, mean cell diameters in the second layer increase dramatically (Figs 11C and 12), indicating cell stretching to compensate for the reduced cell turnover. This phenomenon is analogous to the presence of enlarged cells in the superficial layers of the corneal epithelium. It should be noted that this result does not exclude alternative biological or mechanical mechanisms (e.g., cytoskeletal remodelling, changes in cell-cell adhesion, or differential proliferation rates) that could contribute to this phenotype.

### 4.4 Centripetal velocities of clonal growth

The centripetal velocity, as measured by various experimental studies [8,44,49], is the rate at which the leading edge of a cell clone approaches the centre. To approximate the centripetal velocity of a clonal "spoke" of cells, we chose a particular LESC and its lineage in the basal layer and its corresponding spoke in the second layer to determine the cell velocity.

We then recorded the distance from the circumference of the cell in the clonal lineage cluster that is closest to the centre, which we call the tip (Fig 13). The velocity is obtained by superimposing the tip position in both layers and selecting the greater distance.

The model was used to explore the relationship of shedding rate $r_{shed}$ with the centripetal velocity (Fig 14). Note that the distance of the tip of the clonal spoke from the edge does not always increase. With some non-zero probability, a collection of cells near the tip can shed, causing the tip to retract suddenly. This can be seen in the video animation (S1 Video). The table in Fig 14 shows the velocity of a clonal spoke for various values of $r_{shed}$ found by linear regression fitting.

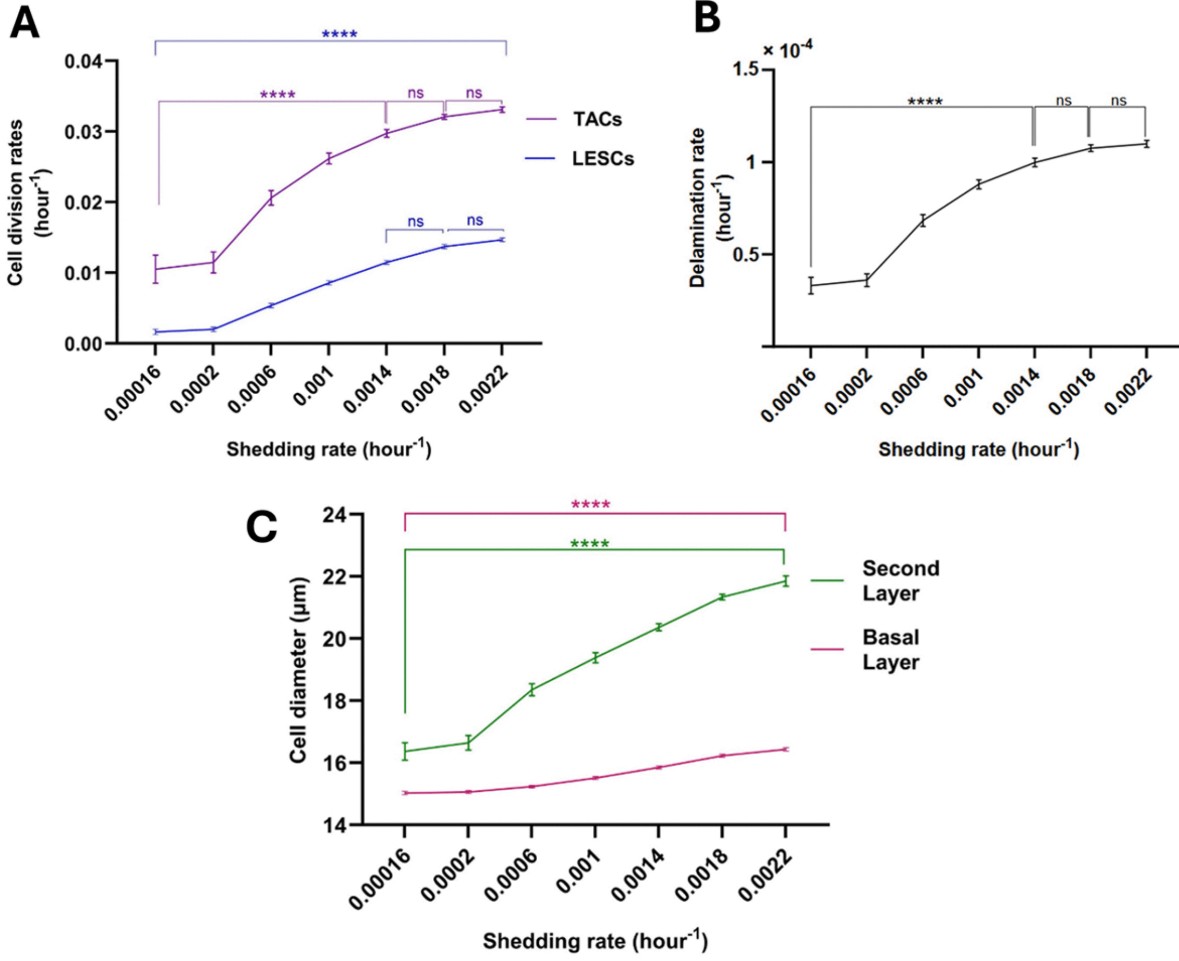

**Fig 11**. Relationship of shedding rate to cell division rates (A) delamination rate (B) and mean cell diameters in the basal and the second layers (C). The mean $\pm$ SD of cell division rates are shown for 20 simulations for each shedding rate. Statistical differences were determined using Kruskal-Wallis test followed by Dunn's multiple comparisons test ("ns" means "not-significant", $^*p < 0.05$, $^{**}p < 0.01$, $^{***}p < 0.001$, $^{****}p < 0.0001$).

Di Girolamo et al. [44] reported that a clonal spoke grows linearly at a rate of 10.8 $\mu$m/day, which is equivalent to 0.45 $\mu$m/hour towards the central cornea. The table in Fig 14 demonstrates a positive correlation between $r_{shed}$ and the velocity of a TAC spoke. In our model, we assume a shedding rate of $r_{shed} = 0.0005$/hour [43], for which the resulting velocity is approximately 0.05 $\mu$m/hour (see Fig 14). This value deviates from the experimentally reported migration rate [44]. However, for $r_{shed} = 0.004$/hour, the corresponding velocity is 0.31 $\mu$m/hour, which is in the ballpark of the reported migration rate [44]. We predict that as the layer three and the higher layers are constructed, the increase in cell size will allow for an enhancement in velocity.

## 5 Sensitivity analysis

To examine the robustness of our model and to identify the parameters (Table 1) that had the greatest influence on various outcomes, we undertook sensitivity analysis for the mean LD in the basal and the second layers, LESC and TAC division rates, number of cells in the basal and the second layers and delamination rate by calculating their SRCCs and p-values (S1 Table).

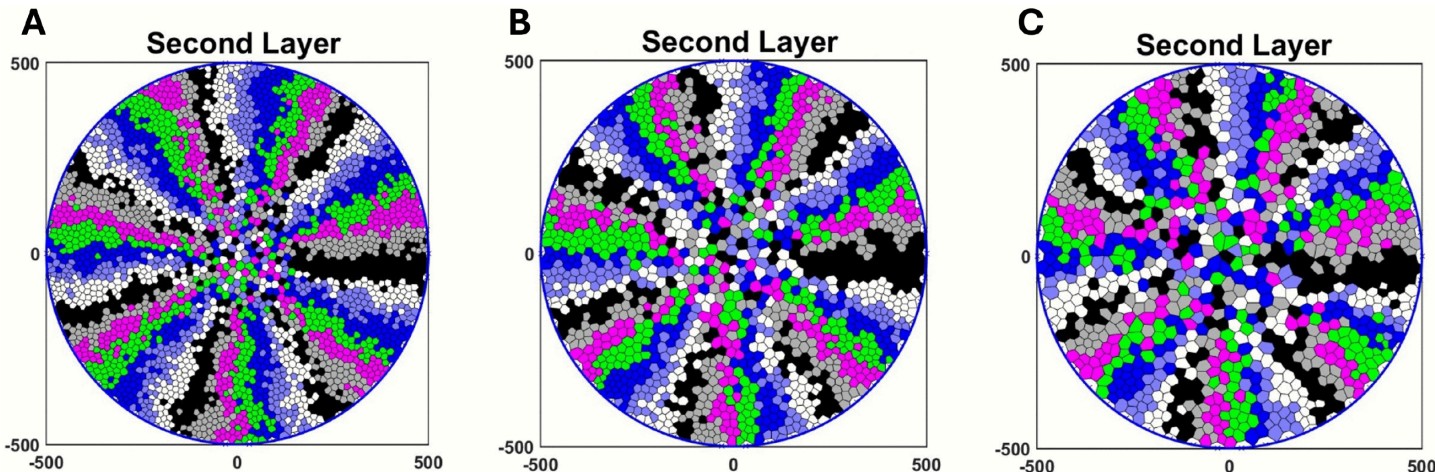

**Fig 12**. The simulation results for the second layer are presented for various values of $r_{shed}$. The findings indicate that as $r_{shed}$ increases, the mean cell diameter also increases and cells undergo stretching to cover the second layer. Specifically, for $r_{shed}$ values of (A) 0.0002/hour, (B) 0.002/hour, and (C) 0.004/hour the mean cell diameters are 17.05 $\mu$m, 22.64 $\mu$m and 26.9 $\mu$m, respectively.

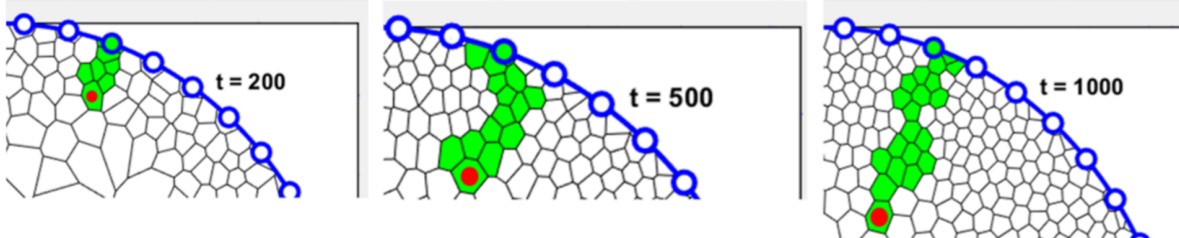

**Fig 13**. A specific TAC spoke, originating from an LESC, is illustrated at various times (200 hour, 500 hour and 1000 hour) with the tip cell indicated by a red dot.

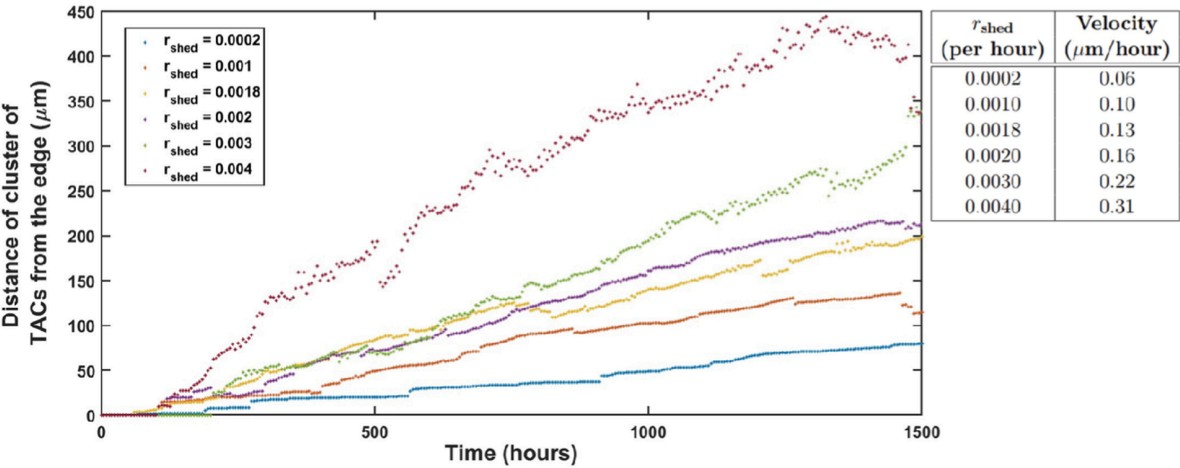

**Fig 14**. **Distance of the tip of a TAC spoke from the edge versus time.** The table presents the velocity of a TAC spoke for various values of $r_{shed}$ over 1500 hours.

As mentioned in Sect 4.1, the LD serves as a measurement of the centripetal nature of cell migration, and a lower LD is associated with more direct centripetal movement. As $n_{max}$ increases, the mean LD increases in the basal layer (Fig 15A), but there is a weak negative correlation between $n_{max}$ and the mean LD in the second layer (Fig 15B). This observation is consistent with our simulations presented in Sect 4.1 and is supported by the results of the Kruskal-Wallis test for both the basal and the second layers (Fig 8).

As the movement parameter $\lambda$ increases, the cell velocity correspondingly increases, resulting in the disruption of the centripetal pattern in both layers. Consequently, a positive correlation is observed between $\lambda$ and the mean LD in both layers (Fig 15A, 15B).

Fig 15C and 15D show that as $n_{max}$ increases, both LESC and TAC division rates decrease. This observation is consistent with our results presented in Fig 9A. Furthermore, increasing the rest length ($s$) leads to an increase in cell size and the cell-substrate adhesion force, which may contribute to a reduction in both delamination and TAC division rates, as shown in Fig 15D. However, there is a positive correlation between the LESC division rate and the rest length.

Additionally, an increase in $F_{LESC}^{th}$ promotes more frequent LESC division, thereby increasing its division rate, while increasing $F_{LESC}^{th}$ decreases TAC division rate, indicating that TACs divide more slowly. In contrast, there is a negative correlation between $F_{TAC}^{th}$ and the LESC division rate, suggesting that an increase in TAC division rate reduces the necessity for LESCs to divide (Fig 15C). Moreover, while increasing $F_{TAC}^{th}$ enhances the TAC division rate, increasing $F_{LESC}^{th}$ has the opposite effect, resulting in a reduced TAC division rate (Fig 15D). Regarding clonal cohesion, increasing both LESC and TAC threshold values leads to a more organized, spoke-like pattern in both layers (Fig 15A, 15B).

LESC and TAC division rates are influenced by $c_{cell-substrate}$. As $c_{cell-substrate}$ increases, there is a reduction in delamination, which subsequently leads to a decreased requirement for cell division to maintain the population among both LESCs (Fig 15C) and TACs (Fig 15D). This raises important and intriguing points, showing how the strength of cell-substrate interactions can affect epithelial tissue turnover without having introduce growth factors.

Furthermore, an increase in $c_{vertical}$, can decrease LD, indicating more clonal cohesion across both layers. In contrast, a decrease in $c_{vertical}$, can increase LD and reduces the spoke-like pattern of migration (Fig 15A, 15B).

The rest length ($s$) plays a critical role in determining the number of cells within the basal layer, as shown in Fig 15E. Specifically, an increase in $s$ leads to a reduction in the number of basal layer cells, corresponding to an increase in individual cell size. Additionally, an increase in $c_{cell-substrate}$ results in a lower delamination rate, thereby increasing the number of cells in the basal layer. However, due to the reduced rate of delamination, the number of cells in the second layer decreases (Fig 15F). In contrast, increasing $c_{neigh}$ leads to a suppression of cell division, resulting in a decrease in the number of cells in both the basal and the second layers. Finally, as the shedding rate ($r_{shed}$) increases, the average cell size in the second layer becomes larger, while the effect on cell size in the basal layer remains relatively minor (Fig 11C).

Fig 15G demonstrates a negative correlation between $n_{max}$ and delamination rate, which is consistent with our findings in Fig 9B. An increase in the rest length ($s$) results in an enlargement of cell size and an enhancement of the cell-substrate adhesion force, which collectively contribute to a reduction in the delamination rate. Furthermore, an increase in $F_{TAC}^{th}$, promotes more TAC divisions, which in turn leads to a higher delamination rate. This indicates a positive correlation between $F_{TAC}^{th}$ and the delamination rate. In contrast, there appears to be a weaker correlation between $F_{LESC}^{th}$ and the delamination rate, indicating that the increase in TAC division may serve to counterbalance cell loss due to delamination, as observed in Fig 10A. Finally, we observe a positive correlation between the shedding rate ($r_{shed}$) and the delamination rate, as illustrated in Fig 11B.

## 6 Discussion

The corneal epithelium, the outermost layer of the cornea, is stratified and consists of 5 to 7 layers of basal, suprabasal, and superficial cells. Through the process of cell delamination, cells are pushed outward from their current layer to the

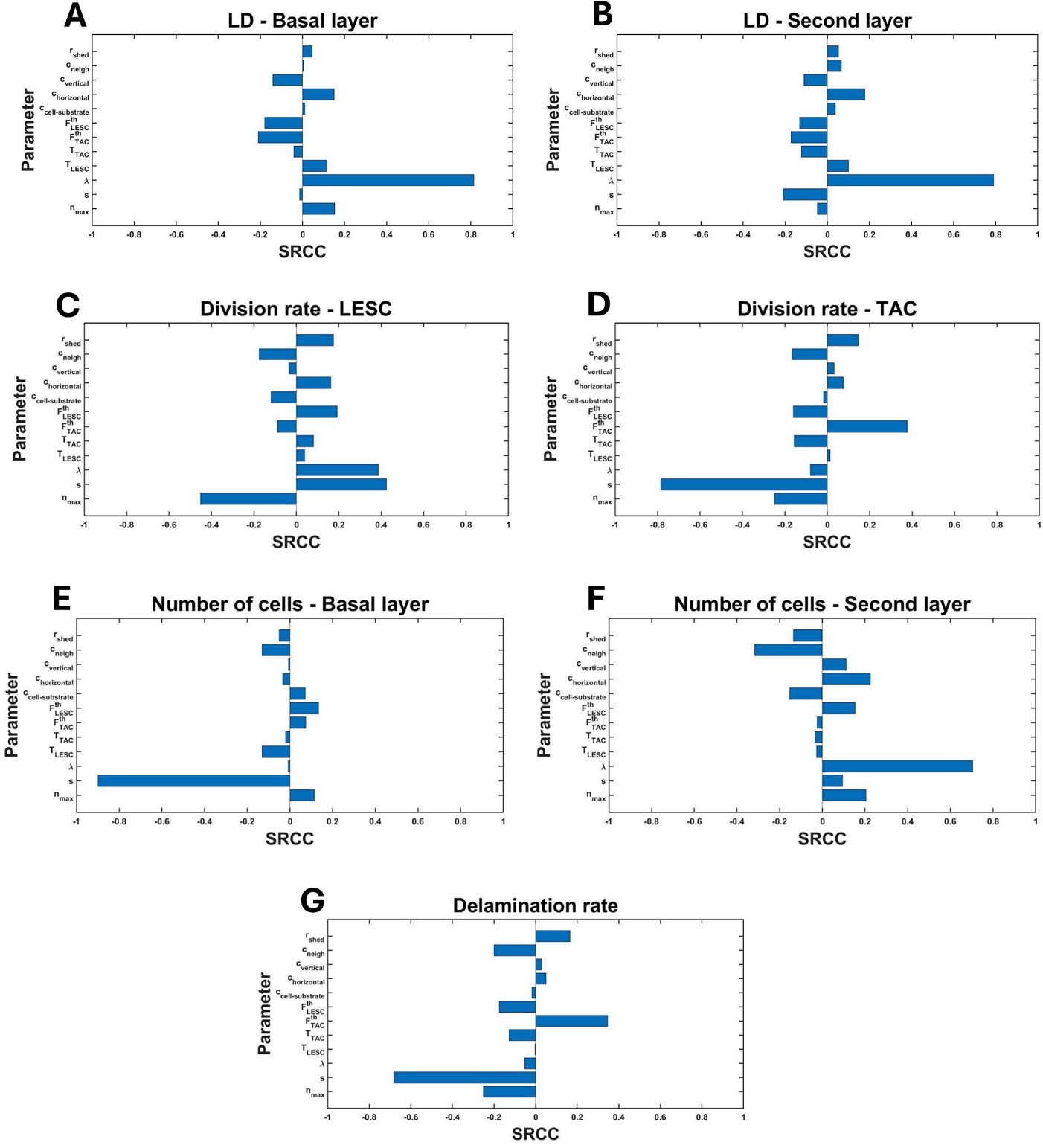

**Fig 15**. Spearman ranked correlation coefficients (SRCC) between parameters (Table 1) and (A) the mean LD in the basal layer, (B) the mean LD in the second layers, (C) LESC division rate, (D) TAC division rate, (E) number of cells in the basal layer, (F) number of cells in the second layer and (G) delamination rate.

next, facilitating stratification. Our two-layer model incorporates delamination, enabling us to capture the essence of epithelial stratification and how communication of forces within and between layers can regulate population balance.

It is well established that the corneal epithelial cells exhibit centripetal migration within their layer, alongside vertical movement to the upper layers. In 2016, Lobo et al. [22] demonstrated through mathematical and biological models that cells in the basal layer migrate centripetally without needing to provide any biological cues such as chemoattractants. Despite this understanding, key questions remain regarding how the layers are organized under normal and wound healing conditions in a 3-dimensional setting. To investigate this, we developed a mathematical model designed as a Voronoi cell-based framework, which integrates two layers of the corneal epithelium. The VCBM function as a valuable visualization tool for biologists in interpreting complex cellular dynamics.

The key differences between our model and the one presented in [22] are as follows. First, the most significant distinction is that our model comprises two layers—the basal layer and the second layer. The addition of a second layer above the basal layer necessitates the inclusion of force transmission between layers to accurately determine delamination, which introduces substantial complexities. In this regard, our model explicitly represents cell delamination as a process governed by three regulatory forces (cell-substrate adhesion, horizontal and vertical forces). Second, the proliferation rate in our model is an emergent feature, rather than being arbitrarily imposed, as in [22]. Third, incorporating friction into the cell movement model to create a more realistic model of motility. Finally, our model incorporates a shedding rate, which governs the removal of cells from the second layer, and we analyze its role in cell dynamics.

By incorporating 2 cell types (LESCs, TACs) and forces, our model can provide us with insight into the importance of various biological parameters in regulating emergent behaviors such as clonal growth rate, cell turnover and cohesiveness of cell lineages.

Our simulations and the sensitivity analysis results (Fig 15) indicated that the centripetal movement in the basal layer extends to the second layer (Fig 7). In agreement with Lobo et al. [22] , the spoke-like pattern in the basal layer is sensitive to $n_{max}$. As this parameter increases, the integrity of this pattern diminishes. In contrast, the pattern in the second layer remains surprisingly unaffected by changes in $n_{max}$. This finding needs further investigation to understand the independence of the organizational structure of the second layer to $n_{max}$. Two properties unique to the basal layer, which might contribute to this distinct behaviour, are adhesion to the basement membrane and the cell division. To explore this further, we can construct more layers above the second layer to determine whether they are also independent of $n_{max}$ or not.

Our next finding pertains to the division rate of LESCs and TACs and their changes in relation to $n_{max}$. Our results indicate that as $n_{max}$ increases, the division rate of TACs and LESCs decrease (Fig 9). Actually, although a lower value of $n_{max}$ resulted in more well-organized corneal structures compared to a higher value of $n_{max}$, it required more frequent divisions of LESCs to sustain population homeostasis.

In the cell division part of the model, we assumed two different minimum cell cycle times for LESCs and TACs ($T_{LESC} = 48(h)$, $T_{TAC} = 24(h)$), and Fig 9 illustrates that TACs exhibit a higher division rate compared to LESCs. This observation may raise concerns that the model is inherently biased due to the assumption that $T_{LESC}$ is twice as big as $T_{TAC}$. To address this, we considered a special case in which both cell types have the same minimum cell cycle duration, $T_{LESC} = T_{TAC} = 24(h)$. Fig 16A shows the cell division rates when we have different and identical minimum cell cycle times. Notably, the absence of significant differences in cell division rates across these cases suggests that the disparity in LESCs and TACs division rates emerges as an intrinsic property of the model rather than being an imposed assumption.

Furthermore, Fig 16B presents the reported division rates of LESCs and TACs obtained from [27]. These estimates are consistent with experimentally measured values, indicating division times of approximately 8 days for LESCs and 4 days for TACs.

Finally, we studied the impact of increased delamination from the basal layer to the second layer on the division rate of LESCs and TACs, by varying the strength of cell interaction with the basement membrane substrate. A modest increase

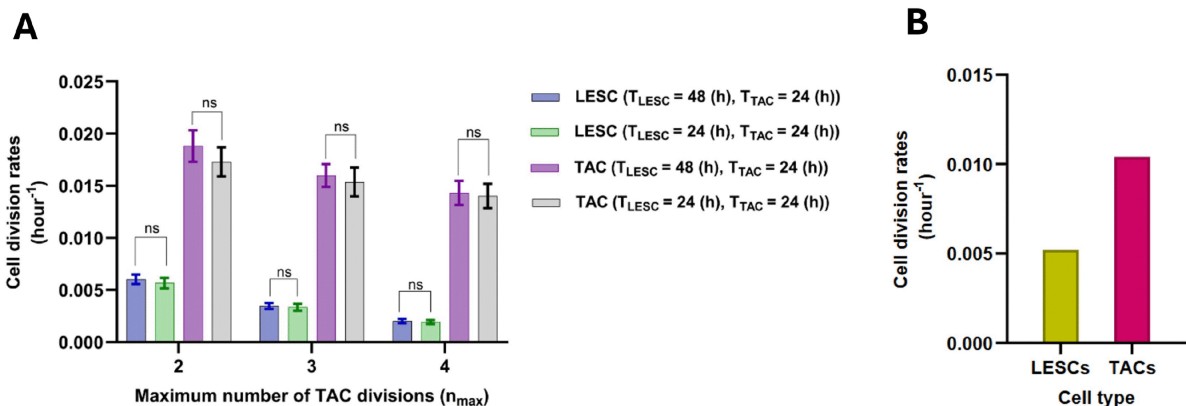

**Fig 16.** (A) Relationship of cell division rates to $n_{max}$ for different and identical minimum cell cycle times. The mean $\pm$ SD of cell division rates are shown for 20 simulations for each $n_{max}$. Statistical differences were determined using Kruskal-Wallis test followed by Dunn's multiple comparisons test ("ns" means "not-significant"). (B) Reported division rates of LESCs and TACs in [27].

in delamination from the baseline value did not produce significant changes in the division rate of LESCs and TACs. However, increases exceeding 20% led to enhanced cycling of both LESCs and TACs (Fig 10), while the division rate of LESCs increases to a lower extent. This suggests that processes that weaken the cell-substrate interaction will cause an increased turnover of epithelial cells without the need to induce growth factors-an observation of particular interest. Notably, exposure of the cornea to ultraviolet radiation leads to both degradation of the basement membrane [6] and enhanced corneal epithelial turnover [22], supporting our finding. Furthermore, this observation suggests that TACs have a buffering effect on LESCs and contribute to their low resting division rate. It means that TACs can regulate the effects of changing conditions (increasing delamination) on LESCs, thereby maintaining a balance in the cell population dynamics.

Several limitations of this model should be considered. Accurately determining several parameters in our model is challenging, particularly given the difficulties to accurately access and measure physical magnitudes at the single cell level for the basal layer of stratified epithelia, including the presence of suprabasal layers and interconnectedness of cells. For this reason, certain parameters were calibrated phenomenologically to reproduce key features of epithelial behaviour. This step is essential to test whether the model can reproduce observed biological behaviours. If the model fails to do so having parameters that can be freely adjusted, it suggests the mechanistic assumptions are incomplete or incorrect. Conversely, the fact that parameter values could be found for our model to successfully reproduce observed phenomena supports that the mechanistic principles embedded in the model are key drivers of the biological process. Further experimental progress and calibration will enable the development of more accurate and predictive models, and a deeper assessment of the wider implications of phenomenologically tuned parameters, a limitation of this model that needs to be considered.

This model is constructed based on parameter values derived from the mouse cornea. We expect similar qualitative behaviour in the human cornea; however, there is a lack of quantitative data for human corneal parameters. Moreover, simulating the human cornea is computationally more expensive due to its larger surface area and the likely greater number of LESCs distributed along the limbal rim.

Another important limitation is that the framework currently is based on two layers of the epithelium — the basal and the second layers. To achieve a more comprehensive understanding of the corneal epithelium and its dynamics, it will be essential to extend the model by incorporating upper layers and investigating their mechanisms. We expect that the

horizontal and vertical forces would be transmitted to the upper layer; however, to fully capture the stratification, an inter-layer adhesive force proportional to the cell area should be introduce.

However, modelling the upper layers, particularly those that are not directly connected to the basal layer, presents a significant challenge in this approach. In this context, several questions arise, such as the relationship between the shedding rate from the uppermost layer and the cell division rate in the basal layer. Furthermore, as cells progress towards the upper layers, their size increases, which may influence the rest length under delamination conditions. Our model currently simulates a healthy tissue in homeostasis. In future with a generalized model, complexity can be added to the model to address pathological situations more realistically, such as wounding of the central cornea or limbus, genetic conditions that affect LESC lifespan and interactions with the stroma via growth factors and other effector molecules.

## Acknowledgments

All Computations were carried out in MATLAB. All statistical analysis was performed with GraphPad Prism10 (Graph-Pad Software Inc., CA, USA). The authors acknowledge the facilities, and the scientific and technical assistance of the Sydney Informatics Hub at the University of Sydney and, in particular, access to the high performance computing facility Artemis.

## Supporting information

**S1 Fig. Simulation of a life-size mouse cornea.** The simulation result for both layers are presented using a corneal radius of 1,500 $\mu$m and 1,000 LESCs, consistent with experimentally observed values reported in [45,47]. The simulation was run for a duration of 8,000 hours from the empty layers with $n_{\max} = 3$. As illustrated, the centripetal movement is preserved in both layers.
(TIFF)

**S2 Fig. Simulation of the centripetal growth in the basal and the second layer for $n_{\mathbf{max}} = 1, 5$.** Simulation of the centripetal growth in the basal and the second layer for (A) $n_{\max} = 1$, (B) $n_{\max} = 5$ to see how the centripetal pattern changes for the extreme values.
(TIFF)

**S1 Video. Clonal growth in the basal and the second layer.** This video animation illustrates the centripetal velocities associated with clonal growth in both layers for $n_{\max} = 3$, $r_{shed} = 0.004$/hour. Notably, the distance between the tip of the clonal spoke and the edge does not consistently increase over time. This is due to occasional cell loss near the tip, which can result in a sudden retraction of the clonal front. One frame represents 20 time steps.
(MP4)

**S1 Table. Sensitivity analysis table.** S1 Table shows SRCC and *p*-values between model parameters and (a) LD - Basal layer, (b) LD - second layer, (c) Division rate - LESC, (d) division rate - TAC, (e) Number of cells - Basal layer, (f) Number of cells - Second layer and (g) delamination rate.
(TIFF)

**S1 Appendix. ODD protocol of the mathematical model.** The structure of this ODD is based on the framework proposed by Grimm et al. [50] for describing agent-based model.
(PDF)

## Author contributions

**Conceptualization:** Neda Khodabakhsh Joniani, Peter S. Kim, J. Guy Lyons.

**Formal analysis:** Neda Khodabakhsh Joniani, David Martinez-Martin, Peter S. Kim.

**Funding acquisition:** Peter S. Kim.

**Investigation:** Neda Khodabakhsh Joniani, Peter S. Kim.

**Methodology:** Neda Khodabakhsh Joniani, David Martinez-Martin, Peter S. Kim.

**Project administration:** Peter S. Kim, J. Guy Lyons.

**Resources:** Peter S. Kim, J. Guy Lyons.

**Software:** Neda Khodabakhsh Joniani, Peter S. Kim.

**Supervision:** Peter S. Kim, J. Guy Lyons.

**Validation:** Neda Khodabakhsh Joniani, David Martinez-Martin, Peter S. Kim, J. Guy Lyons.

**Visualization:** Neda Khodabakhsh Joniani.

**Writing – original draft:** Neda Khodabakhsh Joniani.

**Writing – review & editing:** Neda Khodabakhsh Joniani, David Martinez-Martin, Peter S. Kim, J. Guy Lyons.

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
