## [Decision Letter · Decision Letter 0]

20 Oct 2025

PCOMPBIOL-D-25-01308

Intercellular forces driving stratification in a two-layer corneal epithelium: Insight from a Voronoi cell-based simulation model

PLOS Computational Biology

Dear Dr. Khodabakhsh Joniani,

Thank you for submitting your manuscript to PLOS Computational Biology. After careful consideration, we feel that it has merit but does not fully meet PLOS Computational Biology's publication criteria as it currently stands. Therefore, we invite you to submit a revised version of the manuscript that addresses the points raised during the review process.

Please submit your revised manuscript within 60 days Dec 20 2025 11:59PM. If you will need more time than this to complete your revisions, please reply to this message or contact the journal office at ploscompbiol@plos.org. Please include the following items when submitting your revised manuscript:

We look forward to receiving your revised manuscript.

Kind regards,

Virginia E. Pitzer, Sc.D.

Editor-in-Chief

PLOS Computational Biology

Virginia Pitzer

Editor-in-Chief

PLOS Computational Biology

**Additional Editor Comments (if provided):**

Overall, the reviewers find the modeling framework promising, but they have some questions and suggestions for improving the experimental grounding and visualization of the results.

**Journal Requirements:**

**Reviewers' comments:**

Reviewer's Responses to Questions

**Comments to the Authors:**

Reviewer #1: In the paper, authors developed an interesting mathematical individual-based model that successfully capture stratification driven by intercellular forces in two layers of 2D domain (corneal epithelium). Authors well summarized what has been done in various types of mathematical models based on experimental observation. Each model has the positive and negative aspects in describing the observed behavior. It is quite interesting to see the control and emergence of patters in the upper layer from the lower layer. It is also quite interesting modeling framework that allows the description of patterns in two separate domains. The manuscript was well written and made the points. I think the paper addressed an important issue in corneal epithelium dynamics and it is well worth publishing the paper for PLoS Computation Biology. However, I have a few questions below before it can be accepted. Thus, I would recommend the major revision at the moment.

Major Questions.

1. What happens if there is some damage in the base layer? Would be the system adaptive enough to generate the similar patterns even in response to these perturbed environment?

2. What is the effect of changes in c_vertical in vertical force in generating the patterns?

3. In the model, the cell division occurs under two conditions. And, one of the conditions were the maximum neighbor force is less than a threshold F^{th} for LESCs and TACs. And those two threshold values are different for LESCs and TACs. How were those parameter chosen? I understand that the authors stated in Section 3.4 that two threshold values (F_LESC^th, F_TAC^th) were chosen to obstain the desired dynamics and sensitivity analysis was performed for the effect of different values. However, cell division is a critical step of the whole process and modeling. And, I wonder what are the critical dynamical changes authors expect if different set of the thersholds were chosen? Any thoughts on this?

4. It would be great to see time courses of spatial distributions of cells from different lineages and patterns in both layers instead of showing the snapshot of the pattern at final time? (at t=5,000 hours). This way, we can better understand the whole dynamics.

Minor comments

i) Page 5, 2nd line from Eq. (1): There is a typo in formula: vector F_i^V (here the location of arrow is misplaced and the notation is not consistent with vector F_i^V in Eq. (1)

Reviewer #2: In this manuscript, the author explains the self-renewing ability of cornea using the migrating ability of TACs from LESCs and delamination between these two layers. The author adopts a XYZ hypothesis approach to explain this mechanism, where X means cell proliferation, Y means cell migration and Z means old cell removal. The author uses a 2-dimensional Voronoi cell-based model to explain the cornea epithelial stratification which reveals that there is a direct link between cell loss and cell growth.

There are minor changes needed and some recommendations to improve the quality of this manuscript.

1. There is no specified method section in the manuscript that explain how the simulation was performed.

2. Has the author tried to increase the dt = 5 to dt=8 or 10 to see the difference in the results?

3. Also, you have taken the mouse cornea in consideration for this simulation, how would this model and simulation fit for a human eye cornea? Or has the author tried different parameters for human cornea?

4. Another great addition to the paper would be a time-lapse video of the simulation to see how the cells move and replace the shedding of old cells.

Reviewer #3: Summary

This manuscript introduces a two-layer Voronoi cell-based model to study corneal epithelial stratification via cell proliferation, delamination, and mechanical interactions. The authors extend earlier one-layer models by incorporating vertical forces, substrate adhesion, and shedding, uncovering links between mechanical cues and epithelial turnover. The work is mathematically well-structured and conceptually appealing, particularly the finding that weakening cell–substrate adhesion enhances turnover independent of growth factors.

However, the model is only loosely tied to biological data, several parameters lack justification, and conclusions are sometimes overstated. Strengthening validation, clarifying assumptions, and expanding analysis to better connect with epithelial biology will be essential before publication.

Major Comments

1. Biological Validation and Experimental Comparisons

o The model’s predictions (e.g., enlarged superficial cells under high shedding, TAC buffering of LESCs) are compared qualitatively to literature observations but lack systematic quantitative validation.

o Recommendation: Integrate published data on clone velocities, cell size distributions, or division rates for at least one key parameter (e.g., centripetal velocity vs. shedding rate in Fig. 14). Explicitly state how model outputs align—or diverge—from experiments.

2. Parameter Justification and Sensitivity

o Many parameters (e.g., force constants, thresholds, λ) are chosen to “produce desired dynamics” rather than based on measurements, risking circular logic.

o Recommendation: Provide literature-supported ranges for at least some parameters (e.g., traction forces from corneal epithelial cells). Where data are lacking, highlight model predictions as testable hypotheses rather than fixed assumptions.

3. Biological Scope and Realism

o Cell death, biochemical regulation, and signaling cues are excluded, limiting relevance for real tissue behavior.

o Recommendation: Discuss how growth factors (e.g., EGF, integrins) or apoptosis could be incorporated. At minimum, acknowledge this limitation when interpreting wound-healing dynamics.

4. Overinterpretation of Results

o Statements such as “our model explains the presence of enlarged cells” are too strong; the model recapitulates this phenotype but does not rule out alternative mechanisms (e.g., cytoskeletal remodeling).

o Recommendation: Temper conclusions and clarify that this is a minimal mechanical model rather than a comprehensive explanation.

5. nmax and Division Rate Analysis

o The focus on extreme nmax values (1 and 5) is only loosely biologically motivated since experimental estimates cluster near 2–4.

o Recommendation: Emphasize results within the physiological range; move extreme cases to supplemental data.

6. Figures and Data Presentation

o Several figures (e.g., Figs. 9–11) rely on static bar plots. Emergent dynamics (spoke formation, cell size evolution) are not fully visualized.

o Recommendation: Add time-lapse visualizations, lineage trajectory plots, or size heatmaps. Overlay model predictions with experimental images where possible.

7. Extension to Multilayer Corneal Epithelium

o The real cornea has 5–7 layers; the two-layer model captures only a subset of processes.

o Recommendation: Reframe claims as a foundational step toward multilayer modeling. Outline how additional layers could be incorporated and what new questions this would address.

**Have the authors made all data and (if applicable) computational code underlying the findings in their manuscript fully available?**

Reviewer #1: Yes

Reviewer #2: Yes

Reviewer #3: Yes

PLOS authors have the option to publish the peer review history of their article (what does this mean?). If published, this will include your full peer review and any attached files.

Reviewer #1: No

Reviewer #2: No

Reviewer #3: No

**Figure resubmission:**
---

## [Decision Letter · Decision Letter 1]

2 Feb 2026

Dear Mrs. Khodabakhsh Joniani,

We are pleased to inform you that your manuscript '

Intercellular forces driving stratification in a two-layer corneal epithelium: Insight from a Voronoi cell-based simulation model' has been provisionally accepted for publication in PLOS Computational Biology.

Best regards,

Virginia E. Pitzer, Sc.D.

Editor-in-Chief

PLOS Computational Biology

Virginia Pitzer

Editor-in-Chief

PLOS Computational Biology

Reviewer's Responses to Questions

**Comments to the Authors:**

Reviewer #1: Authors responded and revised in response to all of my questions. The paper can be accepted.

Reviewer #2: All the questions have been addressed.

Reviewer #3: The authors have addressed the major comments and improved the paper significantly. This manuscript is well written, methodologically sound, and makes a clear and important contribution to the field. The conclusions are fully supported by the data, and I have no further substantive concerns. I recommend acceptance as it is.

**Have the authors made all data and (if applicable) computational code underlying the findings in their manuscript fully available?**

Reviewer #1: Yes

Reviewer #2: Yes

Reviewer #3: Yes

PLOS authors have the option to publish the peer review history of their article (what does this mean?). If published, this will include your full peer review and any attached files.

Reviewer #1: No

Reviewer #2: No

Reviewer #3: No

---

## [Editor Report · Acceptance letter]

PCOMPBIOL-D-25-01308R1

Intercellular forces driving stratification in a two-layer corneal epithelium: Insight from a Voronoi cell-based simulation model

Dear Dr Khodabakhsh Joniani,

I am pleased to inform you that your manuscript has been formally accepted for publication in PLOS Computational Biology. Your manuscript is now with our production department and you will be notified of the publication date in due course.

With kind regards,

Judit Kozma
